# Intraoperative Facial Nerve Monitoring during Parotidectomy: The Current Practices and Patterns of the Korean Society of Head and Neck Surgery (KSHNS)

**DOI:** 10.3390/diagnostics14202277

**Published:** 2024-10-13

**Authors:** Dongbin Ahn, Ji Hye Kwak, Geun-Jeon Kim, Heejin Kim, Dong Won Lee, Kwang Jae Cho

**Affiliations:** 1Department of Otolaryngology-Head and Neck Surgery, School of Medicine, Kyungpook National University, Daegu 41566, Republic of Korea; godlikeu@naver.com (D.A.); laugh112@naver.com (J.H.K.); 2Department of Otolaryngology-Head and Neck Surgery, College of Medicine, The Catholic University of Korea, Seoul 06591, Republic of Korea; emelenciana@naver.com; 3Department of Otolaryngology-Head and Neck Surgery, Hallym University Sacred Heart Hospital, Anyang 14068, Republic of Korea; heejin5020@daum.net; 4Department of Otolaryngology-Head and Neck Surgery, School of Medicine, Catholic University of Daegu, Daegu 42472, Republic of Korea; ldw3878@hanmail.net

**Keywords:** parotid, facial nerve, palsy, electromyography, monitoring

## Abstract

**Objectives:** This study aimed to evaluate the current practices and trends of intraoperative facial nerve (FN) monitoring (IOFNM) during parotidectomy. **Methods:** A questionnaire containing 33 questions collecting information on the usage, indications, settings, techniques, loss of signal (LOS) management, anesthesiologist cooperation, and perception of usefulness of IOFNM was distributed among 348 members of the Korean Society of Head and Neck Surgery (KSHNS) via a dedicated website. **Results:** The response rate was approximately 25.6%, and 97% of the respondents reported using IOFNM selectively or routinely during parotidectomy. IOFNM usage decreased as the surgeon’s level of experience increased (*p* = 0.089), from 100% in those with less than 5 years of experience to 75% in those with 20 or more years. Approximately 95% of respondents reported that the initial event threshold for electromyography activity used was 50–149 μV. Moreover, 52.4% of respondents performed neural mapping of the FN before visual identification. Initial management of LOS in visually intact FNs included checking the IOFNM system (75.3%), confirmation of muscle relaxant dosage (75.3%), and facial twitch identification (58.8%). Further management included proceeding with surgery regardless of persistent LOS (81.2%) and steroid administration sometimes or all of the time (72.9%). Overall, 98.8% of respondents found IOFNM beneficial for safe execution of parotidectomy. **Conclusions:** The majority of KSHNS surgeons used IOFNM during parotidectomy, although the clinical implementation of the procedure and LOS management varied between practitioners. This could be attributed to the lack of standardized protocols for IOFNM, emphasizing the need for the development of evidence-based consensus guidelines for all institutions.

## 1. Introduction

The primary principles of parotidectomy include complete tumor removal and preservation of the facial nerve (FN). However, despite extensive efforts by head and neck surgeons to identify and preserve the structural and functional integrity of the FN during parotidectomy, the incidence rates of transient and permanent facial palsy after surgery remain between 9% and 66% and between 0% and 9%, respectively [1,2,3,4,5]. Major injury to the FN can have devastating consequences that result in a significant decrease in the patient’s quality of life. Recent evidence suggests that the frequency of parotidectomy has been steadily increasing, potentially due to the rising incidence of both benign and malignant tumors—particularly Warthin’s tumors—and is associated with increasing life expectancy and advancements in diagnostic techniques [6,7,8,9,10]. Consequently, the clinical implications of parotidectomy and FN management have expanded in the field of head and neck surgery.

Initial intraoperative FN monitoring (IOFNM), which involves direct visualization of facial muscle movements induced by FN stimulation, was first introduced in 1898 to facilitate the identification and preservation of the FN during parotidectomy [11,12,13]. Since then, the application of this technique has been significantly refined with the introduction of electromyography (EMG) in 1970 [12,13,14]. Consistent advances in intraoperative nerve monitoring (IONM) equipment and techniques over the past few decades have led to an increase in the popularity of IOFNM, making it a valuable component of parotidectomy today.

Although there are several international guidelines and consensus statements on the clinical use of IONM for the recurrent laryngeal nerve (i.e., IORLNM), no such standardized protocols on the use and interpretation of IOFNM have been published to date [15,16,17,18,19]. Consequently, the application of IOFNM has been based on numerous indications and techniques that vary with the preferences of the surgeons and the institutions [4,5,20,21,22].

Therefore, this study evaluated the current practices and trends of IOFNM usage in parotidectomies, with the aim of informing the development of an evidence-based standardized protocol for IOFNM.

## 2. Materials and Methods

The Institutional Review Board of Kyungpook National University Hospital approved to waive informed consent based on the methodology of the present study.

Following an extensive review of the existing literature on the clinical applications of IOFNM, the Committee of Nerve Monitoring, the Korean Society of Head and Neck Surgery (KSHNS), developed an online survey containing 33 questions collecting information on the use, indications, settings, techniques, loss of signal (LOS) management, cooperation with anesthesiologists, and perception of usefulness of the technique. An email containing a cover letter providing information on the survey was distributed among members of the KSHNS, and those who provided consent were invited to visit a website where they could complete the survey.

The outcome measures of interest were (1) variations in IOFNM usage by the surgeon’s level of experience and the number of parotidectomies performed, and (2) presumed incidence of FN paralysis based on the frequency of IOFNM usage. All statistical analyses, including the Chi-squared test and Fisher’s exact test, were performed using SPSS (version 18.0; SPSS Inc., Chicago, IL, USA). Statistical significance was defined as a two-sided *p*-value < 0.05.

## 3. Results

### 3.1. Clinical Settings and the Level of Experience of the Respondents

A total of 348 anonymous questionnaires were distributed via the official KSHNS e-mails, of which 89 surveys were completed, resulting in an overall response rate of 25.6%. Of these, 5 (5.6%), 2 (2.3%), and 82 (92.1%) respondents had practiced in private, hospital-based, and academic settings, respectively, while 10 (11.2%), 20 (22.5%), 38 (42.7%), and 21 (23.6%) respondents had <5 years, 5–10 years, 10–20 years, and ≥20 years of experience, respectively. Furthermore, 88 (98.9%) respondents were currently performing parotidectomies, with 18 (20.2%), 27 (30.3%), 24 (27.0%), and 4 (4.4%) respondents completing <20, 20–49, 50–99, and ≥100 parotidectomies per year, respectively (Table 1).

### 3.2. Usage, Indications, and Purpose

Of the 88 respondents who currently performed parotidectomies, 3 (3.4%) reported not using IOFNM due to a lack of equipment (*n* = 2) and the associated increase in the time and cost burden (*n* = 1). Fourteen (15.9%) respondents used IOFNM in selected cases only, with the indications for use including revision (*n* = 13), suspected FN adhesion (*n* = 13), malignancy (*n* = 12), and deep lobe location (*n* = 9), while seventy-one (80.7%) respondents stated that they used IOFNM routinely in all parotidectomies. The reasons for using IOFNM, as reported by the 85 respondents who used IOFNM selectively or routinely, included the prevention of inadvertent FN injury (*n* = 79, 92.9%), identification and mapping of the FN (*n* = 61, 71.8%), intraoperative assessment of FN function (*n* = 28, 32.9%), the education of trainees (*n* = 17, 20.0%), and the prevention of medico-legal problems (*n* = 1, 1.2%) (Table 2).

### 3.3. Trends of IOFNM Usage by Level of Experience and Number of Parotidectomy Procedures Performed

Among the respondents with <5 years, 5–9 years, 10–19.9 years, and ≥20 years of experience, 100% (10/10), 85.0% (17/20), 76.3% (29/38), and 75.0% (15/20) reported routine usage of IOFMN, respectively, indicating a decreasing trend of usage with increasing levels of experience. This trend exhibited a moderate linear-by-linear association (*p* = 0.089; Figure 1). Furthermore, among respondents who performed <20, 20–49, 50–99, and ≥100 parotidectomies per year, 73.7% (14/19), 86.2% (25/29), 79.4% (27/34), and 83.3% (5/6) reported routine usage of IOFMN, respectively, although no linear-by-linear association was observed for this trend (*p*-value = 0.712; Figure 2).

### 3.4. IOFNM Settings and Techniques

Table 3 summarizes the settings in which IOFNM was carried out and the techniques used. Of the 85 respondents who performed IOFNM during parotidectomy, 52 (61.2%) and 33 (38.8%) respondents reported using two-channel and four-channel recording methods with the nerve integrity monitor system, respectively (Figure 3). None of the respondents used surface electrodes for IOFNM. Thirty-seven (43.5%), forty-four (51.8%), three (3.5%), and one (1.2%) respondent reported that the initial event thresholds for EMG activity used were 50–99 μV, 100–149 μV, 150–199 μV, and ≥200 μV, respectively.

A total of 53 (52.4%) respondents used neural mapping of the FN before visual identification of its main trunk, with 2 (3.8%), 26 (49.1%), 14 (26.4%), 6 (11.3%), and 5 (9.4%) respondents using stimulation intensities of 0.5–0.9 mA, 1–1.4 mA, 1.5–1.9 mA, 2.0–2.4 mA, and ≥2.5 mA, respectively, for this procedure. Thirty-two (37.6%) respondents did not use neural mapping prior to visual identification of the FN main trunk. Moreover, 49 (57.6%), 24 (28.2%), 5 (6.0%), and 0 (0.0%) respondents reported using stimulation intensities of 0.5–0.9 mA, 1–1.4 mA, 1.5–1.9 mA, and ≥2.0 mA, respectively, for the FN branches after visual identification. Seven (8.2%) respondents did not stimulate the FN branches as long as the nerve could be identified visually.

To assess the functional integrity of the FN after tumor dissection, 73 (85.9%) respondents reported checking the EMG signal only, 3 (3.5%) reported comparing the initial and final stimulation thresholds, and 5 (6.0%) reported comparing the initial and final maximum response amplitudes. Four (4.6%) respondents stated that they did not check FN function using IOFNM if the nerve appeared intact visually.

### 3.5. Management of LOS

Table 4 summarizes LOS management and the respondents’ experiences of inadvertent FN injury due to false LOS. The initial actions taken upon observation of LOS in visually intact FN branches included checking the IOFNM system (*n* = 64, 75.3%), confirmation of the timing and dosage of the muscle relaxant (*n* = 64, 75.3%), visual and tactile identification of facial twitch during stimulation (*n* = 50, 58.8%), and exploration of adjacent regions to identify other FN branches (*n* = 50, 58.8%). One (1.2%) respondent stated that he would not take any action if the FN appeared intact visually.

The next steps of action upon observation of persistent LOS despite initial management included proceeding with surgery regardless of LOS (*n* = 69, 81.2%), re-stimulation using a higher intensity or lower event threshold (*n* = 34, 40.0%), re-stimulation after waiting for 20–30 min (*n* = 15, 17.6%), and re-stimulation after administration of muscle relaxant antagonists (*n* = 13, 15.3%). None of the respondents discontinued the surgery because of persistent LOS. Upon observation of true LOS, 45 (52.9%) and 17 (20.0%) respondents reported using steroid administration sometimes or all of the time, respectively. Twenty-three (27.1%) respondents never used steroids for LOS. Thirty (35.3%) and two (2.4%) respondents reported encountering transient and permanent FN injury due to false LOS, respectively.

### 3.6. Anesthetic Considerations When Using IOFNM

Further analysis showed that 40 (51.7%), 36 (42.3%), and 5 (6.0%) respondents reported high, moderate, and low levels of cooperation among the institution’s anesthesiologists during IOFNM, respectively. Furthermore, 24 (28.2%) respondents actively asked for a reduction in the muscle relaxant dosage by informing the anesthesiologist during IOFNM, while 61 respondents left dose reduction to the anesthesiologist’s discretion. Seventeen (20.0%) respondents confirmed the exact dose of the muscle relaxant administered, while sixty-eight (80.0%) respondents did not.

### 3.7. The Surgeons’ Perceptions of the Usefulness of IOFNM

Table 5 summarizes the surgeons’ perceptions of the usefulness of IOFNM. Sixty-six (77.6%) respondents stated that IOFNM was useful for preventing both transient and permanent FN injury, five (6.0%) respondents stated that it was useful for preventing permanent FN injuries only, and thirteen (15.2%) respondents found it useful in selected cases (i.e., revision, deep lobes, and malignancies) only. Overall, 84 (98.8%) respondents stated that IOFNM was beneficial when performing parotidectomy, while 1 (1.2%) respondent found it not beneficial. Eighty-one (95.3%) respondents stated that they will use IOFNM if they undergo parotidectomy as a patient.

### 3.8. Presumed Incidence of FN Injury during Parotidectomy

The presumed incidence of transient FN injury during parotidectomy was <5.0%, 5%–9.9%, 10%–19.9%, 20%–29.9%, and ≥30.0% among 56 (65.9%), 22 (25.9%), 6 (7.0%), 1 (1.2%), and 0 (0.0%) respondents, respectively. The presumed incidence of permanent FN injury during parotidectomy was <1.0%, 1–4.9%, and ≥5.0% among 71 (83.5%), 14 (16.5%), and 0 (0.0%) respondents, respectively. The distribution of the presumed incidence of transient and permanent FN injury did not differ with selective or routine usage of IOFNM (Table 6).

## 4. Discussion

The current survey-based study showed that the majority of surgeons who were members of the KSHNS used IOFNM when performing parotidectomies and considered it beneficial for the prevention of transient and permanent FN injury. However, the practical implementation of IOFNM and the management of LOS varied between these surgeons.

Overall, 96.6% of respondents in the current study reported using IOFNM, and this proportion was higher than that reported previously by studies conducted in Germany (75%), the UK (82%), and the USA (60%) [20,21,22]. However, these previous studies were published in 2005–2006 and, therefore, may not accurately reflect current rates of usage in these countries. A more recent survey study conducted in Spain in 2021 found that IOFNM was used in approximately 94% of parotidectomies, and these findings were comparable to those observed in the current study, thus emphasizing the important role of IOFNM as a component of parotidectomy [23].

Approximately 80% of respondents in the current study used IOFNM routinely, although this rate was seen to decrease as the surgeon’s level of experience increased. This could potentially be attributed to improvements in the surgeon’s confidence and technical skills with increasing surgical experience in parotidectomies. In fact, comprehensive knowledge of the nerve anatomy and meticulous surgical techniques are the best ways to prevent iatrogenic FN injury. IOFNM is not mandatory and, instead, represents an ancillary technique that facilitates precise and safe execution of parotidectomy. However, it is also associated with an increased burden of resources required (e.g., device, time, cost, and technique). Therefore, while novice surgeons who are still gaining familiarity with the nerve anatomy may find routine usage of IOFNM valuable, it is likely to offer limited benefits for veteran surgeons who are highly proficient in performing parotidectomies [24].

The practical implementation of IOFNM was relatively heterogeneous among the survey respondents, particularly with regard to neural mapping and the stimulation intensities used. Although there are no standardized protocols for IOFNM to date, Chiang et al. recently proposed a method of standardization wherein 5 mA was used for localization of the FN main trunk and 3–5 mA was used for pre- and post-dissection FN stimulation [4]. However, this stimulation intensity could potentially prove to be risky and cause harm to the nerve, particularly since dedicated FN monitors typically limit their maximum stimulus outputs to 3–5 mA [25]. Moreover, the normal FN threshold in the extracranial region ranges between 0.1 and 0.3 mA, while an injured FN may require higher intensities of 1.0–2.0 mA to elicit an EMG signal [26]. In clinical practice, direct stimulation of the FN at 0.5–1 mA is common, as is the use of 1–3 mA to briefly map out a region where the FN may be located [26,27,28]. Given those facts, the findings of the current study showed that the majority of surgeons used acceptable stimulation intensities, although a certain degree of heterogeneity between respondents was observed. Moreover, 92.9% of surgeons included in the current study used 50–150 μV as a signal threshold, and this was in agreement with previous studies that reported using 50 or 100 μV as a signal threshold for IOFNM [26,27,28,29].

In the current study, the majority of surgeons (85.9%) reported that they evaluated the functional integrity of the FN after tumor dissection by checking EMG signal generation only and did not take the stimulation threshold, amplitude, latency, and number of events into consideration. This disregard of IOFNM parameters can potentially be attributed to the lack of a well-defined, widely accepted IOFNM interpretation protocol and a lack of consensus on the correlation between the information obtained from the major parameters of IOFNM and postoperative FN palsy. A previous study conducted in 2016 and involving 25 patients undergoing parotidectomy with IOFNM showed that none of the parameters (i.e., stimulation threshold, EMG amplitude, and latency) effectively predicted possible FN dysfunction after surgery [30]. However, another study conducted in 2019 and involving 222 patients undergoing parotid surgery with IOFNM reported that the optimal cut-off values for the stimulation threshold and number of mechanical events were 0.25 mA and 8, respectively. These findings suggest that thresholds >0.25 mA combined with >8 mechanical events were associated with a 77% risk of postoperative nerve weakness [29]. Another study conducted in 2021 and involving 112 patients reported that an amplitude decrease >50% in an FN branch was associated with an increased incidence of facial dysfunction [31]. Therefore, the inconsistencies in the evidence and recommendations reported to date emphasize the need for the establishment of a standardized interpretation protocol for IOFNM that enables surgeons to preserve postoperative FN function using the key parameters, thereby promoting the usefulness of the procedure further.

While LOS in IORLNM has been clearly defined as <100μV by the International Neural Monitoring Study Group in 2011, no such definition exists for IOFNM [32]. Consequently, no practical guidelines on the management of LOS during IOFNM exist, and this is in contrast to IORLNM, where appropriate protocols have been established and described extensively. In the current study, initial management of LOS for visually intact FN branches included checking the IOFNM system (75.3%), confirmation of the muscle relaxant dosage (75.3%), identification of facial muscle twitch (58.8%), and exploration of other FN branches (58.8%). These actions were similar to the recommendations (e.g., checking the monitoring system, observation of laryngeal twitch, etc.) included in the LOS management guidelines for IORLNM [15,16,32]. However, 37.7% of respondents in the current study stated that they had encountered FN injury due to false LOS, highlighting the need for standardized management guidelines for IOFNM that help surgeons distinguish between true and false LOSs and provide appropriate management where necessary.

In the current study, 72.9% of surgeons managed LOS by administering steroids sometimes or all of the time. However, there is a dearth of clinical evidence on the efficacy of steroids for LOS management in IOFNM, although they have been used to reduce neuronal edema and decrease the consequent risk of neurapraxia dysfunction in numerous surgical procedures [33,34,35,36]. Evidence on the administration of steroids for LOS during ORLNM is also insufficient and conflicting, with one prospective study involving 115 patients with LOS during thyroidectomy reporting that postoperative vocal fold paralysis did not diminish following intraoperative administration of steroids [37]. However, two recent studies reported promising results that suggested that administration of a single dose of steroids after LOS can promote signal recovery compared with observation [38,39]. Therefore, the role of intraoperative steroids in the management of LOS during IOFNM or IORLNM should be investigated further to facilitate the development of an evidence-based protocol.

To date, no prospective randomized controlled clinical trials have examined the efficacy of IOFNM in minimizing FN morbidity during parotidectomy, preventing the development of a better understanding of the true benefits of the procedure. A systematic review and meta-analysis conducted in 2015 and including 546 patients from seven studies found that the incidence of permanent facial palsy did not differ significantly between the IOFNM and unmonitored groups (3.9% vs. 7.1%; *p*-value = 0.18), while the incidence of immediate facial palsy was seen to be significantly lower in the IOFNM group (22.5% vs. 34.9%; *p*-value = 0.001) [13]. A more recent systematic review and meta-analysis conducted in 2021 and including 1069 patients from 10 studies found that the incidence of immediate and permanent FN palsy following parotidectomy was significantly lower in the IOFNM group compared to the unmonitored group [23.4% vs. 38.4% (*p*-value = 0.001) and 5.7% vs. 13.6% (*p*-value = 0.001), respectively] when all studies were included, although this difference was seen to disappear upon inclusion of prospective studies only [5]. Despite the uncertainties regarding the true efficacy of IOFNM in reducing the risk of FN injury, the majority of surgeons who used the technique considered it to be beneficial for safe execution of parotidectomy and the prevention of FN injury. Therefore, further prospective randomized controlled clinical trials are necessary to demonstrate the true benefits of IOFNM by reducing the gap between the belief in clinical practice and evidence from research outcomes.

This study has several limitations. First, this study was conducted exclusively among Korean surgeons, which may limit the generalizability of the findings to other regions or international surgical practices. Second, as the present study relies on self-reported data from a survey, there is potential for response bias, where surgeons’ reported practices may not fully reflect their actual clinical behaviors. Additionally, this study focuses on reported practices rather than direct clinical outcomes, meaning it does not directly assess the effectiveness of IOFNM in reducing facial nerve injury. Despite these limitations, this study provides valuable clinical insights by offering a comprehensive overview of current practices regarding IOFNM during parotidectomy. This information can serve as a foundation for developing evidence-based consensus guidelines and addressing the need for standardized IOFNM protocols.

## 5. Conclusions

This study demonstrated that the majority of surgeons in the KSHNS used IOFNM during parotidectomy, although its use decreased as the surgeons’ level of experience increased. Additionally, significant heterogeneity was observed among surgeons regarding the clinical application of the procedure and the management of LOS, possibly due to the lack of a standardized protocol for IOFNM. In order to maximize the use of IOFNM, consensus guidelines on the indications for use, device settings, stimulation techniques, parameter interpretation, and management of LOS should be established. The findings of this study provide valuable insights into the current practice of IOFNM which can serve as the basis for future research in this field.

## Figures and Tables

**Figure 1 diagnostics-14-02277-f001:**
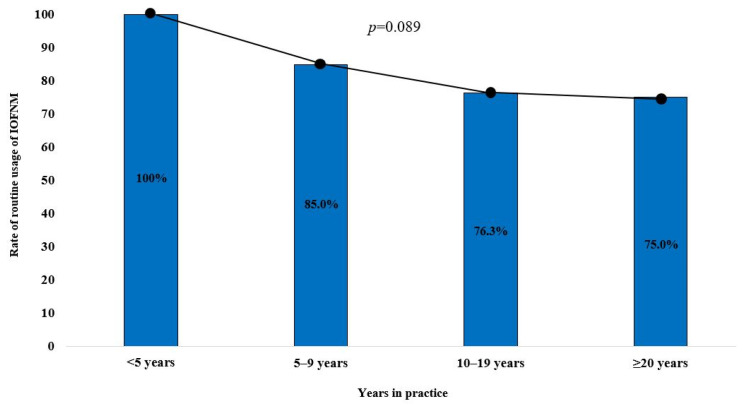
The distribution of routine usage of IOFMN by the surgeons’ level of experience in clinical practice.

**Figure 2 diagnostics-14-02277-f002:**
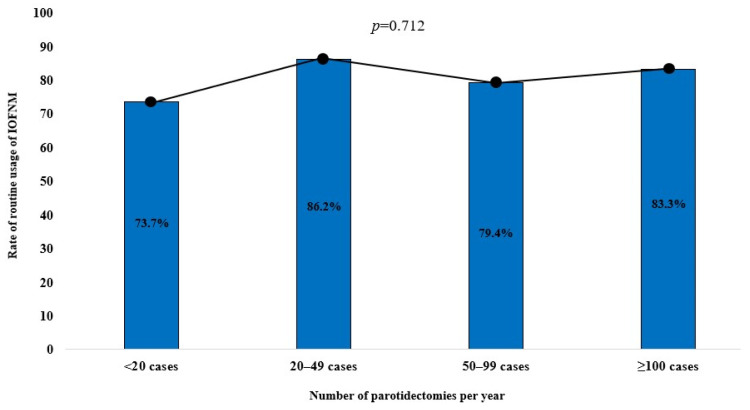
The distribution of routine usage of IOFMN by the number of parotidectomies performed per year.

**Figure 3 diagnostics-14-02277-f003:**
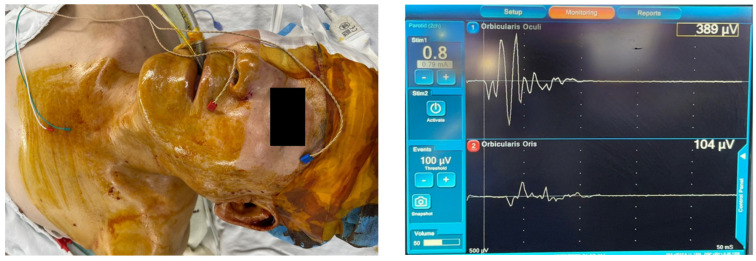
The setup of intraoperative facial nerve monitoring (IOFNM) using a 2-channel needle electrode. Two pairs of needle electrodes are inserted deeply into the orbicularis oculi (blue) and orbicularis oris (red) muscles (**Left**). The initial stimulation intensity (Stim1) and the event thresholds for EMG activity (Events) are set to 0.8 mA and 100 μV, respectively. The two elicited EMG signals, representing the function of the relevant facial nerve branches, are displayed on the monitoring screen (**Right**).

**Table 1 diagnostics-14-02277-t001:** Clinicodemographic characteristics.

	Respondents (*N* = 89)
Clinical setting	
Private	5 (5.6%)
Hospital-based	2 (2.3%)
Academic	82 (92.1%)
Years in practice	
<5 years	10 (11.2%)
5–10 years	20 (22.5%)
10–20 years	38 (42.7%)
≥20 years	21 (23.6%)
Number of parotidectomies per year	
None	1 (1.1%)
<20	19 (21.3%)
20–49	29 (32.5%)
50–99	34 (38.2%)
≥100	6 (6.7%)

**Table 2 diagnostics-14-02277-t002:** Usage, indications, and purpose of intraoperative facial nerve monitoring (IOFNM).

	Respondents (*n* = 88)
Type of IOFNM usage	
Never	3 (3.4%)
due to no equipment	2
due to time and cost burden	1
Selective	14 (15.9%)
Personal indications *	
Revision	13
Possible adhesion	13
Malignancy	12
Deep lobe location	9
Retrograde dissection	4
Large tumor	3
Routine	71 (80.7%)
Purpose of using IOFNM * (for 85 IOFNM users)	
Prevention of inadvertent FN injury	79 (92.9%)
Facilitation of identification and mapping FN	61 (71.8%)
Intraoperative assessment of FN function	28 (62.9%)
Education	17 (20.0%)
Prevention of medico-legal issues	1 (1.2%)

* Multiple responses were included. IOFNM, intraoperative facial nerve monitoring; FN, facial nerve.

**Table 3 diagnostics-14-02277-t003:** IOFNM settings and techniques.

	Respondents (*n* = 85)
Channel and electrode	
2-channel recording with needle electrode	52 (61.2%)
4-channel recording with needle electrode	33 (38.8%)
2- or 4-channel recording with surface electrode	0 (0.0%)
Event threshold setting	
50–99 μV	37 (43.5%)
100–149 μV	44 (51.8%)
150–199 μV	3 (3.5%)
≥200 μV	1 (1.2%)
Neural mapping of FN before visual identification of FN main trunk	
No	32 (37.6%)
Yes	53 (62.4%)
Stimulation intensity	
0.5–0.9 mA	2
1–1.4 mA	26
1.5–1.9 mA	14
2.0–2.4 mA	6
≥2.5 mA	5
Assessment of initial FN function after visual identification of FN	
No	7 (8.2%)
Yes	78 (91.8%)
Stimulation intensity	
0.5–0.9 mA	49
1–1.4 mA	24
1.5–1.9 mA	5
≥2.0 mA	0
Assessment of final FN function after tumor dissection	
No	4 (4.6%)
Yes	81 (95.4%)
Technique	
Check generation of EMG signal	73
Compare initial and final stimulation thresholds	3
Compare initial and final maximal response amplitudes	5

FN, facial nerve.

**Table 4 diagnostics-14-02277-t004:** Management of loss of signal (LOS).

	Respondents (*n* = 85)
Initial action for LOS *	
Checking the IOFNM system	64 (75.3%)
Confirmation of the timing and dosage of the muscle relaxant	64 (75.3%)
Visual and tactile identification of facial twitch during stimulation	50 (58.8%)
Exploration of adjacent regions to identify other FN branches	50 (58.8%)
None	1 (1.2%)
Management of persistent LOS after initial action *	
Proceeding with surgery	69 (81.2%)
Re-stimulation using a higher intensity or lower event threshold	34 (40.0%)
Re-stimulation after waiting for 20–30 min	15 (17.6%)
Re-stimulation after administration of muscle relaxant antagonists	13 (15.3%)
Discontinuing surgery	0 (0.0%)
Administration of steroid for true LOS	
Never	23 (27.1%)
Sometimes	45 (52.9%)
All the time	17 (20.0%)
Inadvertent FN injury due to false LOS	
No	53 (62.4%)
Yes	32 (37.6%)
Transient injury	30
Permanent injury	2

* Multiple responses were included. FN, facial nerve.

**Table 5 diagnostics-14-02277-t005:** The surgeons’ perceptions of the usefulness of IOFNM.

	Respondents (*n* = 85)
Perception of usefulness of IOFNM	
Useful for preventing both transient and permanent FN injury	66 (77.6%)
Useful for preventing transient FN injury only	0 (0.0%)
Useful for preventing permanent FN injury only	5 (6.0%)
Useful in selective cases	13 (15.2%)
Not useful	1 (1.2%)
Is IOFNM generally beneficial for safety parotidectomy?	
No	1 (1.2%)
Yes	84 (98.9%)
Would you use IONM if you undergo parotidectomy as a patient?	
No	1 (1.2%)
Yes	81 (95.3%)
Uncertain	3 (3.5%)

FN, facial nerve.

**Table 6 diagnostics-14-02277-t006:** Presumed incidence of FN injury during parotidectomy according to usage type.

	Selective Use (*n* = 14)	Routine Use (*n* = 71)	*p*-Value
Transient FN injury			
<5.0%	11 (78.6%)	45 (63.4%)	0.590
5.0–9.9%	3 (21.4%)	19 (26.8%)	
10.0–19.9%	0 (0.0%)	6 (8.5%)	
20.0–29.9%	0 (0.0%)	1 (1.4%)	
Permanent FN injury			
<1.0%	13 (92.9%)	58 (81.7%)	0.448
1.0–4.9%	1 (7.1%)	13 (18.3%)	

FN, facial nerve.

## Data Availability

The data that support the findings of this study are available from the corresponding author upon reasonable request.

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
