# Peer review of "Intraoperative Facial Nerve Monitoring during Parotidectomy: The Current Practices and Patterns of the Korean Society of Head and Neck Surgery (KSHNS)"

_diagnostics, 2024, doi:10.3390/diagnostics14202277_

Round 1
Reviewer 1 Report
Comments and Suggestions for Authors
Dear authors,
I found your manuscript very interesting. To further enhance the quality of the manuscript, please find my comments down below:
- I suggest updating the title to incorporate the complete term for KSHNS (Korean Society of Head and Neck Surgery), along with the abbreviated form in parentheses. Consistency is crucial, therefore every abbreviation in the manuscript should follow the same approach.
- The abstract needs to provide more information by including the key findings and their corresponding p-values. This will improve the clearness and effectiveness of the main findings of the research.
- Think about including clinical photographs in the Introduction or Discussion sections to provide a clearer demonstration of the intraoperative facial nerve monitoring technique.
- Separating participants into two groups according to their experience, such as clinicians with various years of practice ( less and more than 10 years) and those with different numbers of parotidectomies performed (more than and less than 50), could be advantageous. These groups could be compared with other parameters or survey responses to identify any notable variations in practice trends.
- It is crucial to acknowledge the study's limitations and strengths at the conclusion of the Discussion section.
- I recommend adding the separated Conclusions section and highlighting key discoveries in bullet points.
- Please make sure the manuscript follows the citation guidelines of the journal. It is important to carefully review how references are cited in the text to guarantee consistency and accuracy.
Kind regards!
Comments on the Quality of English LanguageThe manuscript would benefit from a minor grammar and spell check to enhance readability and refine the text.
Author Response
We wish to resubmit our manuscript titled “Intraoperative Facial Nerve Monitoring during Parotidectomy: Current Practices and Patterns of the Korean Society of Head and Neck Surgery (KSHNS),” which I request you to consider for publication as a Special Issue in the Diagnostics. The manuscript ID is diagnostics-3217556.
We thank the editors and reviewers for their meticulous review of our manuscript. We have revised it according to their comments and suggestions. The point-by-point responses to the comments are appended below, and the revisions are denoted in blue font in the revised manuscript. In addition, the revised manuscript has been edited by a professional English editing company to improve the language, flow, and readability.
We hope that the revised manuscript is now suitable for publication in your esteemed journal. Thank you for your consideration. I look forward to hearing from you.
Point 1. I suggest updating the title to incorporate the complete term for KSHNS (Korean Society of Head and Neck Surgery), along with the abbreviated form in parentheses. Consistency is crucial, therefore every abbreviation in the manuscript should follow the same approach.
Response. Thank you for your valuable comments. I have revised the title in accordance with your suggestion. Additionally, I have reviewed and updated all abbreviations throughout the manuscript for consistency.
Point 2. The abstract needs to provide more information by including the key findings and their corresponding p-values. This will improve the clearness and effectiveness of the main findings of the research.
Response. Thank you for your insightful comments. I have revised the abstract to include more detailed information regarding the use of IOFNM, incorporating specific results and corresponding p-values.
Point 3. Think about including clinical photographs in the Introduction or Discussion sections to provide a clearer demonstration of the intraoperative facial nerve monitoring technique.
Response. Thank you for your helpful suggestion. We have included clinical photographs of IOFNM using a 2-channel needle electrode, which have been added as Figure 3 in the revised manuscript to better illustrate the IOFNM technique for the readers.
Point 4. Separating participants into two groups according to their experience, such as clinicians with various years of practice ( less and more than 10 years) and those with different numbers of parotidectomies performed (more than and less than 50), could be advantageous. These groups could be compared with other parameters or survey responses to identify any notable variations in practice trends.
Response. Thank you for your comments. In the original manuscript, we compared the usage of IOFNM based on surgeons’ experience level and the number of parotidectomies performed per year. Additionally, we compared the presumed incidence of facial nerve injury during parotidectomy according to the type of usage (routine vs. selective). Given the limitations inherent in survey-based studies, we were unable to perform further comparisons that would yield meaningful findings. We appreciate your understanding of this limitation
Point 5. It is crucial to acknowledge the study's limitations and strengths at the conclusion of the Discussion section.
Response. Thank you for your thoughtful suggestion. We have added a section discussing the limitations and strengths of the study in the revised manuscript’s Discussion section.
Point 6. I recommend adding the separated Conclusions section and highlighting key discoveries in bullet points.
Response. Thank you for your suggestion. We have created a separate Conclusions section, highlighting the key findings of the study.
Point 7. Please make sure the manuscript follows the citation guidelines of the journal. It is important to carefully review how references are cited in the text to guarantee consistency and accuracy.
Response. Thank you for your helpful comments. We have revised the reference style in the manuscript to comply with the journal's citation guidelines.
Reviewer 2 Report
Comments and Suggestions for Authors
This is an interesting article that deals with a usefulness of intraoperative facial nerve monitoring during parotidectomy among surgeons in South Korea. The article presents the usefulness of IOFNM, which is widely used in developed countries during parotidectomy. The article is well written. I would like to add just a few things.
- In the introduction: it is advisable to include information about the increasing incidence of Warthin's tumor (WT), which is mainly localized in the parotid gland. This increase in incidence can be explained by improved diagnostics in oncology patients (PET/CT). Another possible explanation for the increasing incidence of WT may be related to patient obesity and increasing life expectancy.
- Discussion: The risk of facial nerve palsy after parotidectomy may be increased by high amperage and number of stimulations during surgery.
Percentages in Table 3 do not add to 100%.
Author Response
We wish to resubmit our manuscript titled “Intraoperative Facial Nerve Monitoring during Parotidectomy: Current Practices and Patterns of the Korean Society of Head and Neck Surgery (KSHNS),” which I request you to consider for publication as a Special Issue in the Diagnostics. The manuscript ID is diagnostics-3217556.
We thank the editors and reviewers for their meticulous review of our manuscript. We have revised it according to their comments and suggestions. The point-by-point responses to the comments are appended below, and the revisions are denoted in blue font in the revised manuscript. In addition, the revised manuscript has been edited by a professional English editing company to improve the language, flow, and readability.
We hope that the revised manuscript is now suitable for publication in your esteemed journal. Thank you for your consideration. I look forward to hearing from you.
Point 1. In the introduction: it is advisable to include information about the increasing incidence of Warthin's tumor (WT), which is mainly localized in the parotid gland. This increase in incidence can be explained by improved diagnostics in oncology patients (PET/CT). Another possible explanation for the increasing incidence of WT may be related to patient obesity and increasing life expectancy.
Response. Thank you for your valuable comments. We have added information regarding the rising incidence of parotid tumors, particularly Warthin’s tumor, along with potential explanations for this trend in the Introduction section of the revised manuscript.
Point 2. Discussion: The risk of facial nerve palsy after parotidectomy may be increased by high amperage and number of stimulations during surgery.
Response. Thank you for your insightful comments. We completely agree with your perspective. As such, we have already addressed in the manuscript that stimulation intensities of 3–5 mA can pose a risk to the nerve and that the normal facial nerve (FN) threshold in the extracranial region typically ranges between 0.1–0.3 mA.
Point 3. Percentages in Table 3 do not add to 100%.
Response. Thank you for pointing out this error. We have reviewed and corrected the values in Table 3.
Round 2
Reviewer 1 Report
Comments and Suggestions for Authors
Dear authors,
thank you for incorporating my suggestions to your manuscript. In my opinion, now it is significantly enhanced and therefore should be accepted for publication.
Congrats and best regards!